# Ensuring Fairness Beyond the Training Data

**Debmalya Mandal**
dm3557@columbia.edu
Columbia University

**Samuel Deng**
sd3013@columbia.edu
Columbia University

**Suman Jana**
suman@cs.columbia.edu
Columbia University

**Jeannette M. Wing**
wing@columbia.edu
Columbia University

**Daniel Hsu**
djhsu@cs.columbia.edu
Columbia University

## Abstract

We initiate the study of fair classifiers that are robust to perturbations in the training distribution. Despite recent progress, the literature on fairness has largely ignored the design of fair and robust classifiers. In this work, we develop classifiers that are fair not only with respect to the training distribution, but also for a class of distributions that are weighted perturbations of the training samples. We formulate a min-max objective function whose goal is to minimize a distributionally robust training loss, and at the same time, find a classifier that is fair with respect to a class of distributions. We first reduce this problem to finding a fair classifier that is robust with respect to the class of distributions. Based on online learning algorithm, we develop an iterative algorithm that provably converges to such a fair and robust solution. Experiments on standard machine learning fairness datasets suggest that, compared to the state-of-the-art fair classifiers, our classifier retains fairness guarantees and test accuracy for a large class of perturbations on the test set. Furthermore, our experiments show that there is an inherent trade-off between fairness robustness and accuracy of such classifiers.

## 1 Introduction

Machine learning (ML) systems are often used for high-stakes decision-making, including bail decision and credit approval. Often these applications use algorithms trained on past biased data, and such bias is reflected in the eventual decisions made by the algorithms. For example, Bolukbasi et al. [9] show that popular word embeddings implicitly encode societal biases, such as gender norms. Similarly, Buolamwini and Gebru [10] find that several facial recognition softwares perform better on lighter-skinned subjects than on darker-skinned subjects. To mitigate such biases, there have been several approaches in the ML fairness community to design fair classifiers [4, 19, 35].

However, the literature has largely ignored the robustness of such fair classifiers. The "fairness" of such classifiers are often evaluated on the sampled datasets, and are often unreliable because of various reasons including biased samples, missing and/or noisy attributes. Moreover, compared to the traditional machine learning setting, these problems are more prevalent in the fairness domain, as the data itself is biased to begin with. As an example, we consider how the optimized pre-processing algorithm [11] performs on ProPublica's COMPAS dataset [1] in terms of *demographic parity* (DP), which measures the difference in accuracy between the protected groups. Figure 1 shows two situations – (1) unweighted training distribution (in blue), and (2) weighted training distributions (in red). The optimized pre-processing algorithm [11] yields a classifier that is almost fair on the unweighted training set (DP $\leq 0.02$). However, it has DP of at least $0.2$ on the weighted set, despite the fact that the marginal distributions of the features look almost the same for the two scenarios.

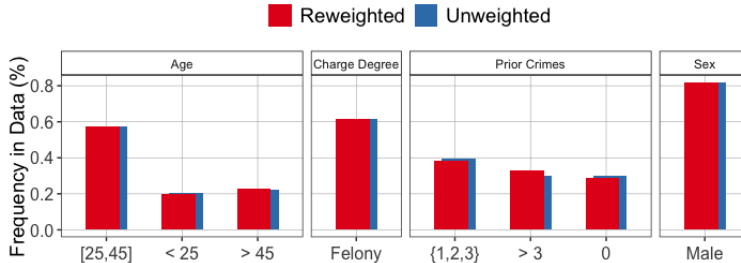

Figure 1: *Unweighted vs Reweighted COMPAS dataset.* The marginals of the two distributions are almost the same, but standard fair classifiers show demographic parity of at least $0.2$ on the reweighted dataset.

This example motivates us to design a fair classifier that is robust to such perturbations. We also show how to construct such weighted examples using a few linear programs.

**Contributions**: In this work, we initiate the study of fair classifiers that are robust to perturbations in the training distribution. The set of perturbed distributions are given by any arbitrary weighted combinations of the training dataset, say $\mathcal{W}$. Our main contributions are the following:

- We develop classifiers that are fair not only with respect to the training distribution, but also for the class of distributions characterized by $\mathcal{W}$. We formulate a min-max objective whose goal is to minimize a distributionally robust training loss, and simultaneously, find a classifier that is fair with respect to the entire class.

- We first reduce this problem to finding a fair classifier that is robust with respect to the class of distributions. Based on online learning algorithm, we develop an iterative algorithm that provably converges to such a fair and robust solution.

- Experiments on standard machine learning fairness datasets suggest that, compared to the state-of-the-art fair classifiers, our classifier retains fairness guarantees and test accuracy for a large class of perturbations on the test set. Furthermore, our experiments show that there is an inherent trade-off between fairness robustness and accuracy of such classifiers.

**Related Work**: Numerous proposals have been laid out to capture bias and discrimination in settings where decisions are delegated to algorithms. Such formalization of fairness can be *statistical* [14, 19–21, 28], *individual* [13, 31], *causal* [23, 25, 36], and even *procedural* [18]. We restrict attention to statistical fairness, which fix a small number of groups in the population and then compare some statistic (e.g., accuracy, false positive rate) across these groups. We mainly consider the notion of *demographic parity* [14, 20, 21] and *equalized odds* [19] in this paper, but our method of designing robust and fair classifiers can be adapted to any type of statistical fairness.

On the other hand, there are three main approaches for designing a fair classifier. The *pre-processing* approach tries to transform training data and leverage standard classifiers [11, 14, 20, 35]. The *in-processing* approach, on the other hand, directly modifies the learning algorithm to meet the fairness criteria [4, 15, 22, 34]. The *post-processing* approach, however, modifies the decisions of a classifier [19, 28] to make it fair. Ours is an in-processing approach and mostly related to [4, 5, 22]. Agarwal et al. [4] and Alabi et al. [5] show how binary classification problem with group fairness constraints can be reduced to a sequence of cost-sensitive classification problems. Kearns et al. [22] follow a similar approach, but instead consider a combinatorial class of subgroup fairness constraints. Recently, [7] integrated and implemented a range of such fair classifiers in a GitHub project, which we leverage in our work.

In terms of technique, our paper falls in the category of *distributionally robust optimization* (DRO), where the goal is to minimize the worst-case training loss for any distribution that is close to the training distribution by some metric. Various types of metrics have been considered including bounded $f$-divergence [8, 26], Wasserstein distance [2, 17], etc. To the best of our knowledge, prior literature has largely ignored enforcing constraints such as fairness in a distributionally robust sense. Further afield, our work has similarity with recent work in fairness testing inspired by the literature on program verification [6, 16, 32]. These papers attempt to automatically discover discrimination in decision-making programs, whereas we develop tools based on linear program to discover distributions that expose potential unfairness.

## 2 Problem and Definitions

We will write $((x, a), y)$ to denote a training instance where $a \in \mathcal{A}$ denotes the protected attributes, $x \in \mathcal{X}$ denotes all the remaining attributes, and $y \in \{0, 1\}$ denotes the outcome label. For a hypothesis $h$, $h(x, a) \in \{0, 1\}$ denotes the outcome predicted by it, on an input $(x, a)$. We assume that the set of hypothesis is given by a class $\mathcal{H}$. Given a loss function $\ell : \{0, 1\} \times \{0, 1\} \rightarrow \mathbb{R}$, the goal of a standard fair classifier is to find a hypothesis $h^* \in \mathcal{H}$ that minimizes the training loss $\sum_{i=1}^n \ell(h(x_i, a_i), y_i)$ and is also fair according to some notion of fairness.

We aim to design classifiers that are fair with respect to a class of distributions that are weighted perturbations of the training distribution. Let $\mathcal{W} = \{w \in \mathbb{R}^n_+ : \sum_i w_i = 1\}$ be the set of all possible weights. For a hypothesis $h$ and weight $w$, we define the *weighted empirical risk*, $\ell(h, w) = \sum_{i=1}^n w_i \ell(h(x_i, a_i), y_i)$. We will write $\delta_F^w(h)$ to define the "unfairness gap" with respect to the weighted empirical distribution defined by the weight $w$ and fairness constraint $F$ (e.g., demographic parity (DP) or equalized odds (EO)). For example, $\delta_{DP}^w(h)$ is defined as

$$\delta_{DP}^w(h) = \max_{a, a' \in \mathcal{A}} \left| \frac{\sum_{i:a_i = a} w_i h(x_i, a)}{\sum_{i:a_i = a} w_i} - \frac{\sum_{i:a_i = a'} w_i h(x_i, a')}{\sum_{i:a_i = a'} w_i} \right|. \tag{1}$$

Therefore, $\delta_{DP}^w(h)$ measures the maximum weighted difference in *acceptance rates* between the two groups with respect to the distribution that assigns weight $w$ to the training examples. On the other hand, $\delta_{EO}^w(h) = 1/2(\delta_{EO}^w(h|0) + \delta_{EO}^2(h|1))$[1], where $\delta_{EO}^w(h|y)$ is defined as

$$\delta_{EO}^w(h|y) = \max_{a, a' \in \mathcal{A}} \left| \frac{\sum_{i:a_i = a, y_i = y} w_i h(x_i, a)}{\sum_{i:a_i = a, y_i = y} w_i} - \frac{\sum_{i:a_i = a', y_i = y} w_i h(x_i, a')}{\sum_{i:a_i = a', y_i = y} w_i} \right|.$$

Therefore, $\delta_{EO}^w(h|0)$ (resp., $\delta_{EO}^w(h|1)$) measures the weighted difference in false (resp., true) positive rates between the two groups with respect to the weight $w$. We will develop our theory using DP as an example of a notion of fairness, but our experimental results will concern both DP and EO.

We are now ready to formally define our main objective. For a class of hypothesis $\mathcal{H}$, let $\mathcal{H}_{\mathcal{W}} = \{h \in \mathcal{H} : \delta_F^w(h) \le \epsilon \ \forall w \in \mathcal{W}\}$ be the set of hypothesis that are $\epsilon$-fair with respect to all the weights in the set $\mathcal{W}$. Our goal is to solve the following min-max problem:

$$\min_{h \in \mathcal{H}_{\mathcal{W}}} \max_{w \in \mathcal{W}} \ell(h, w) \tag{2}$$

Therefore, we aim to minimize a robust loss with respect to a class of distributions indexed by $\mathcal{W}$. Additionally, we also aim to find a classifier that is fair with respect to such perturbations.

We allow our algorithm to output a randomized classifier, i.e., a distribution over the hypothesis $\mathcal{H}$. This is necessary if the space $\mathcal{H}$ is non-convex or if the fairness constraints are such that the set of feasible hypothesis $\mathcal{H}_{\mathcal{W}}$ is non-convex. For a randomized classifier $\mu$, its weighted empirical risk is $\ell(\mu, w) = \sum_h \mu(h)\ell(h, w)$, and its expected unfairness gap is $\delta_F^w(\mu) = \sum_h \mu(h)\delta_F^w(h)$.

## 3 Design

Our algorithm follows a top-down fashion. First we design a meta algorithm that reduces the min-max problem of Equation (2) to a loss minimization problem with respect to a sequence of weight vectors. Then we show how we can design a fair classifier that performs well with respect a fixed weight vector $w \in \mathcal{W}$ in terms of accuracy, but is fair with respect to the entire set of weights $\mathcal{W}$.

### 3.1 Meta Algorithm

Algorithm 1 provides a meta algorithm to solve the min-max optimization problem defined in Equation (2). The algorithm is based on ideas presented in [12], which, given an $\alpha$-approximate Bayesian oracle for distributions over loss functions, provides an $\alpha$-approximate robust solution. The algorithm can be viewed as a two-player zero-sum game between the learner who picks the

**ALGORITHM 1:** Meta-Algorithm
---
**Input:** Training Set: $\{x_i, a_i, y_i\}_{i=1}^n$, set of weights: $\mathcal{W}$, hypothesis class $\mathcal{H}$, parameters $T$ and $\eta$.
Set $\eta = \sqrt{2/T_m}$ and $w_0(i) = 1/n$ for all $i \in [n]$
$h_0 = \text{ApxFair}(w_0)$ /* `Approximate solution of` $\arg\min_{h \in \mathcal{H}_\mathcal{W}} \sum_{i=1}^n \ell(h(x_i, a_i), y_i)$. &emsp; */
**for** *each* $t \in [T_m]$ **do**
&emsp; $w_t = w_{t-1} + \eta \nabla_w \ell(h_{t-1}, w_{t-1})$
&emsp; $w_t = \Pi_\mathcal{W}(w_t)$ /* `Project` $w_t$ `onto the set of weights` $\mathcal{W}$. &emsp; */
&emsp; $h_t = \text{ApxFair}(w_t)$ /* `Approximate solution of` $\min_{h \in \mathcal{H}_\mathcal{W}} \sum_{i=1}^n w_t(i)\ell(h(x_i, a_i), y_i)]$. &emsp; */
**end**
**Output:** $h_f$: Uniform distribution over $\{h_0, h_1, \ldots, h_T\}$.
---

hypothesis $h_t$, and an adversary who picks the weight vector $w_t$. The adversary performs a projected gradient descent every step to compute the best response. On the other hand, the learner solves a fair classification problem to pick a hypothesis which is fair with respect to the weights $\mathcal{W}$ and minimizes weighted empirical risk with respect to the weight $w_t$. However, it is infeasible to compute an exact optima of the problem $\min_{h \in \mathcal{H}_\mathcal{W}} \sum_{i=1}^n w_t(i)\ell(h(x_i, a_i), y_i)]$. So the learner uses an approximate fair classifier $\text{ApxFair}(\cdot)$, which we define next.

**Definition 1.** *ApxFair$(\cdot)$ is an $\alpha$-approximate fair classifier, if for any weight $w \in \mathbb{R}_+^n$, ApxFair$(w)$ returns a hypothesis $\widehat{h}$ such that*

$$\sum_{i=1}^n w_i \ell(\widehat{h}(x_i, a_i), y_i) \leq \min_{h \in \mathcal{H}_\mathcal{W}} \sum_{i=1}^n w_i \ell(h(x_i, a_i), y_i) + \alpha.$$

Using the $\alpha$-approximate fair classifier, we have the following guarantee on the output of Algorithm 1.

**Theorem 1.** *Suppose the loss function $\ell(\cdot, \cdot)$ is convex in its first argument and ApxFair$(\cdot)$ is an $\alpha$-approximate fair classifier. Then, the hypothesis $h_f$, output by Algorithm 1 satisfies*

$$\max_{w \in \mathcal{W}} \mathbb{E}_{h \sim h_f} \left[ \sum_{i=1}^n w_i \ell(h(x_i, a_i), y_i) \right] \leq \min_{h \in \mathcal{H}_\mathcal{W}} \max_{w \in \mathcal{W}} \ell(h, w) + \sqrt{\frac{2}{T_m}} + \alpha$$

The proof uses ideas from [12], except that we use an additive approximate best response.[2]

## 3.2 Approximate Fair Classifier

We now develop an $\alpha$-approximate fair and robust classifier. For the remainder of this subsection, let us assume that the meta algorithm (Algorithm 1) has called the $\text{ApxFair}(\cdot)$ with a weight vector $w^0$ and our goal is to design a classifier that minimizes weighted empirical risk with respect to the weight $w^0$, but is fair with respect to the set of all weights $\mathcal{W}$, i.e., find $f \in \arg\min_{h \in \mathcal{H}_\mathcal{W}} \ell(h, w^0)$. Our method applies the following three steps.

1. Discretize the set of weights $\mathcal{W}$, so that it is sufficient to design an approximate fair classifier with respect to the set of discretized weights. In particular, if we discretize each weight up to a multiplicative error $\epsilon$, then developing an $\alpha$-approximate fair classifier with respect to the discretized weights gives $O(\alpha + \epsilon)$-fair classifier with respect to the set $\mathcal{W}$.

2. Introduce a Lagrangian multiplier for each fairness constraint i.e. for each of the discretized weights, and pair of protected attributes. This lets us set up a two-player zero-sum game for the problem of designing an approximate fair classifier with respect to the set of discretized weights. Here, the learner chooses a hypothesis, whereas an adversary picks the most "unfair" weight in the set of discretized weights.

3. Design a learning algorithm for the learner's learning algorithm, and design an approximate solution to the adversary's best response to the learner's chosen hypothesis. This lets us write an iterative algorithm where at every step, the learner choosed a hypothesis, and the adversary adjusts the Lagrangian multipliers corresponding to the most violating fairness constraints.

We point out that Agarwal et al. [4] was the first to show that the design of a fair classifier can be formulated as a two-player zero-sum game (step 2). However, they only considered group-fairness constraints with respect to the training distribution. The algorithm of Alabi et al. [5] has similar limitations. On the other hand, we consider the design of robust and fair classifier and had to include an additional discretization step (1). Finally, the design of our learning algorithm and the best response oracle is significantly different than [4, 5, 22].

### 3.2.1 Discretization of the Weights

We first discretize the set of weights $\mathcal{W}$ as follows. Divide the interval $[0, 1]$ into buckets $B_0 = [0, \delta)$, $B_{j+1} = [(1 + \gamma_1)^j \delta, (1 + \gamma_1)^{j+1} \delta)$ for $j = 0, 1, \ldots, M - 1$ for $M = \lceil \log_{1+\gamma_1}(1/\delta)) \rceil$. For any weight $w \in \mathcal{W}$, construct a new weight $w' = (w'_1, \ldots, w'_n)$ by setting $w'_i$ to be the upper-end point of the bucket containing $w_i$, for each $i \in [n]$.

We now substitute $\delta = \frac{\gamma_1}{2n}$ and write $\mathcal{N}(\gamma_1, \mathcal{W})$ to denote the set containing all the discretized weights of the set $\mathcal{W}$. The next lemma shows that a fair classifier for the set of weights $\mathcal{N}(\gamma_1, \mathcal{W})$, is also a fair classifier for the set of weights $\mathcal{W}$ up to an error $4\gamma_1$.

**Lemma 1.** *If $\forall w \in \mathcal{N}(\gamma_1, \mathcal{W})$, $\delta^w_{DP}(f) \leq \epsilon$, then we have $\delta^w_{DP}(f) \leq \epsilon + 4\gamma_1$ for any $w \in \mathcal{W}$.*

Therefore, in order to ensure that $\delta^w_{DP}(f) \leq \varepsilon$ we discretize the set of weights $\mathcal{W}$ and enforce $\varepsilon - 4\gamma_1$ fairness for all the weights in the set $\mathcal{N}(\gamma_1, \mathcal{W})$. This result makes our work easier as we need to guarantee fairness with respect to a finite set of weights $\mathcal{N}(\gamma_1, \mathcal{W})$, instead of a large and continuous set of weights $\mathcal{W}$. However, note that, the number of weights in $\mathcal{N}(\gamma_1, \mathcal{W})$ can be $O\left(\log^n_{1+\gamma_1}(2n/\gamma_1)\right)$, which is exponential in $n$. We next see how to avoid this problem.

### 3.3 Setting up a Two-Player Zero-Sum Game

We formulate the problem of designing a fair and robust classifier with respect to the set of weights in $\mathcal{N}(\gamma_1, \mathcal{W})$ as a two-player zero-sum game. Let $R(w, a, f) = \frac{\sum_{i:a_i=a} w_i f(x_i, a)}{\sum_{i:a_i=a} w_i}$. Then $\delta^w_{DP}(f) = \sup_{a, a'} |R(w, a, f) - R(w, a', f)|$. Our aim is to solve the following problem.

$$\min_{h \in \mathcal{H}} \sum_{i=1}^{n} w_i^0 \ell(h(x_i, a_i), y_i) \tag{3}$$
$$\text{s.t. } R(w, a, h) - R(w, a', h) \leq \varepsilon - 4\gamma_1 \ \forall w \in \mathcal{N}(\gamma_1, \mathcal{W}) \ \forall a, a' \in \mathcal{A}$$

We form the following Lagrangian.

$$\min_{h \in \mathcal{H}} \max_{\|\lambda\|_1 \leq B} \sum_{i=1}^{n} w_i^0 \ell(h(x_i, a_i), y_i) + \sum_{w \in \mathcal{N}(\gamma_1, \mathcal{W})} \sum_{a, a' \in \mathcal{A}} \lambda_w^{a, a'} (R(w, a, h) - R(w, a', h) - \varepsilon + 4\gamma_1). \tag{4}$$

Notice that we restrict the $\ell_1$-norm of the Lagrangian multipliers by the parameter $B$. We will later see how to choose this parameter $B$. We first convert the optimization problem define in Equation (4) as a two-player zero-sum game. Here the learner's pure strategy is to play a hypothesis $h$ in $\mathcal{H}$. Given the learner's hypothesis $h \in \mathcal{H}$, the adversary picks the constraint (weight $w$ and groups $a, a'$) that violates fairness the most and sets the corresponding coordinate of $\lambda$ to $B$. Therefore, for a fixed hypothesis $h$, it is sufficient for the adversary to play a vector $\lambda$ such that either all the coordinates of $\lambda$ are zero or exactly one is set to $B$. For such a pair $(h, \lambda)$ of hypothesis and Largangian multipliers, we define the payoff matrix as

$$U(h, \lambda) = \sum_{i=1}^{n} w_i^0 \ell(h(x_i, a_i), y_i) + \sum_{w \in \mathcal{N}(\gamma_1, \mathcal{W})} \sum_{a, a' \in \mathcal{A}} \lambda_w^{a, a'} (R(w, a, h) - R(w, a', h) - \varepsilon + 4\gamma_1)$$

Now our goal is to compute a $\nu$-approximate minimax equilibrium of this game. In the next subsection, we design an algorithm based on online learning. The algorithm uses best responses of the $h$- and $\lambda$-players, which we discuss next.

**Best response of the $h$-player**: For each $i \in [n]$, we introduce the following notation

$$\Delta_i = \sum_{w \in \mathcal{N}(\gamma_1, \mathcal{W})} \sum_{a' \neq a_i} \left( \lambda_w^{a_i, a'} - \lambda_w^{a', a_i} \right) \frac{w_i}{\sum_{j:a_j=a_i} w_j}$$

With this notation, the payoff becomes

$$U(h, \lambda) = \sum_{i=1}^{n} w_i^0 \ell(h(x_i, a_i), y_i) + \Delta_i h(x_i, a_i) - (\varepsilon - 4\gamma_1) \sum_{w \in \mathcal{N}(\varepsilon/5, \mathcal{W})} \sum_{a, a' \in \mathcal{A}} \lambda_w^{a, a'}$$

Let us introduce the following costs.

$$c_i^0 = \begin{cases} \ell(0, 1)w_i^0 & \text{if } y_i = 1 \\ \ell(0, 0)w_i^0 & \text{if } y_i = 0 \end{cases} \qquad c_i^1 = \begin{cases} \ell(1, 1)w_i^0 + \Delta_i & \text{if } y_i = 1 \\ \ell(1, 0)w_i^0 + \Delta_i & \text{if } y_i = 0 \end{cases} \qquad (5)$$

Then the $h$-player's best response becomes the following cost-sensitive classification problem.

$$\widehat{h} \in \underset{h \in \mathcal{H}}{\arg\min} \sum_{i=1}^{n} \left\{ c_i^1 h(x_i, a_i) + c_i^0 (1 - h(x_i, a_i)) \right\} \qquad (6)$$

Therefore, as long as we have access to an oracle for the cost-sensitive classification problem, the $h$-player can compute its best response. Note that, the notion of a cost-sensitive classification as an oracle was also used by [4, 22]. In general, solving this problem is NP-hard, but there are several efficient heuristics that perform well in practice. We provide further details about how we implement this oracle in the section devoted to the experiments.

**Best response of the $\lambda$-player**: Since the fairness constraints depend on the weights non-linearly (e.g., see Eq. (1)), finding the most violating constraint is a non-linear optimization problem. However, we can guess the marginal probabilities over the protected groups. If we are correct, then the most violating weight vector can be found by a linear program. Since we cannot exactly guess this particular value, we instead discretize the set of marginal probabilities, iterate over them, and choose the option with largest violation in fairness.

This intuition can be formalized as follows. We discretize the set of all marginals over $|\mathcal{A}|$ groups by the following rule. First discretize $[0, 1]$ as $0, \delta, (1 + \gamma_2)^j \delta$ for $j = 1, 2, \ldots, M$ for $M = O(\log_{1+\gamma_2}(1/\delta))$. This discretizes $[0, 1]^{\mathcal{A}}$ into $M^{|\mathcal{A}|}$ points, and then retain the discretized marginals whose total sum is at most $1 + \gamma_2$, and discard all other points. Let us denote the set of such marginals as $\Pi(\gamma_2, \mathcal{A})$. Algorithm 2 goes through all the marginals $\pi$ in $\Pi(\gamma_2, \mathcal{A})$ and for each such tuple and a pair of groups $a, a'$ finds the weight $w$ which maximizes $R(w, a, h) - R(w, a', h)$. Note that this can be solved using a linear Program as the weights assigned to a group is fixed by the marginal tuple $\pi$. Out of all the solutions, the algorithm picks the one with the maximum value. Then it checks whether this maximum violates the constraint (i.e., greater than $\varepsilon$). If so, it sets the corresponding $\lambda$ value to $B$ and everything else to 0. Otherwise, it returns the zero vector. As the weight returned by the LP need not correspond to a weight in $\mathcal{N}(\gamma_1, \mathcal{W})$, it rounds the weight to the nearest weight in $\mathcal{N}(\gamma_1, \mathcal{W})$. For discretizing the marginals we will set $\delta = (1 + \gamma_2)\frac{\gamma_1}{n}$, which implies that the number of LPs run by Algorithm 2 is at most $O\left(\log_{1+\gamma_2}^{|\mathcal{A}|}\left(\frac{n}{(1+\gamma_2)\gamma_1}\right)\right) = O(\text{poly}(\log n))$, as the number of groups is fixed.

**Lemma 2.** *Algorithm 2 is an $B(4\gamma_1 + \gamma_2)$-approximate best response for the $\lambda$-player—i.e., for any $h \in \mathcal{H}$, it returns $\lambda^*$ such that $U(h, \lambda^*) \geq \max_\lambda U(h, \lambda) - B(4\gamma_1 + \gamma_2)$.*

**Learning Algorithm**: We now introduce our algorithm for the problem defined in Equation (4). In this algorithm, the $\lambda$-player uses Algorithm 2 to compute an approximate best response, whereas the $h$-player uses Regularized Follow the Leader (RFTL) algorithm [3, 30] as its learning algorithm. RFTL is a classical algorithm for online convex optimization (OCO). In OCO, the decision maker takes a decision $x_t \in \mathcal{K}$ at round $t$, an adversary reveals a convex loss function $f_t : \mathcal{K} \to \mathbb{R}$, and the decision maker suffers a loss of $f_t(x_t)$. The goal is to minimize regret, which is defined as $\max_{u \in \mathcal{K}} \{\sum_{t=1}^{T} f_t(x_t) - f_t(u)\}$, i.e., the difference between the loss suffered by the learner and the best fixed decision. RFTL requires a strongly convex regularization function $R : \mathcal{K} \to \mathbb{R}_{\geq 0}$, and chooses $x_t$ according to the following rule:

$$x_t = \underset{x \in \mathcal{K}}{\arg\min} \, \eta \sum_{s=1}^{t-1} \nabla f_s(x_s)^T x + R(x).$$

We use RFTL in our learning algorithm as follows. We set the regularization function $R(x) = 1/2\|x\|_2^2$, and loss function $f_t(h_t) = U(h_t, \lambda_t)$ where $\lambda_t$ is the approximate best-response to $h_t$.

**ALGORITHM 2:** Best Response of the $\lambda$-player

**Input:** Training Set: $\{x_i, a_i, y_i\}_{i=1}^n$, and hypothesis $h \in \mathcal{H}$.

**for** *each* $\pi \in \Pi(\gamma_2, \mathcal{A})$ *and* $a, a' \in \mathcal{A}$ **do**

> Solve the following LP:
>
> $$w(a, a', \pi) = \arg\max_w \quad \frac{1}{\pi_a} \sum_{i:a_i=a} w_i h(x_i, a) - \frac{1}{\pi_{a'}} \sum_{i:a_i=a'} w_i h(x_i, a')$$
>
> $$\text{s.t.} \quad \sum_{i:a_i=a} w_i = \pi_a \quad \sum_{i:a_i=a'} w_i = \pi_{a'} \quad w_i \geq 0 \quad \forall i \in [n] \quad \sum_{i=1}^n w_i = 1$$
>
> Set $\text{val}(a, a', \pi) = \frac{1}{\pi_a} \sum_{i:a_i=a} w(a, a', \pi)_i h(x_i, a) - \frac{1}{\pi_{a'}} \sum_{i:a_i=a'} w(a, a', \pi)_i h(x_i, a')$

**end**

Set $(a_1^*, a_2^*, \pi^*) = \arg\max_{a,a',\pi} \text{val}(a, a', \pi)$

**if** $val(a_1^*, a_2^*, \pi^*) > \varepsilon$ **then**

> Let $w = w(a_1^*, a_2^*, \pi^*)$.
>
> For each $i \in [n]$, let $w_i'$ be the upper-end point of the bucket containing $w_i$.
>
> **return** $\lambda_w^{a,a'} = \begin{cases} B & \text{if } (a, a', w) = (a_1^*, a_2^*, w') \\ 0 & \text{o.w.} \end{cases}$

**else**

> **return** $\vec{0}$

Therefore, at iteration $t$ the learner needs to solve the following optimization problem.

$$h_t \in \arg\min_{h \in \mathcal{H}} \eta \sum_{s=1}^{t-1} \nabla_{h_s} U(h_s, \lambda_s)^T h + \frac{1}{2}\|h\|_2^2. \tag{7}$$

Here with slight abuse of notation we write $\mathcal{H}$ to include the set of randomized classifiers, so that $h(x_i, a_i)$ is interpreted as the probability that hypothesis $h$ outputs 1 on an input $(x_i, a_i)$. Now we show that the optimization problem (Eq. (7)) can be solved as a cost-sensitive classification problem. For a given $\lambda_s$, the best response of the learner is the following:

$$\widehat{h} \in \arg\min_{h \in \mathcal{H}} \sum_{i=1}^n c_i^1(\lambda_s) h(x_i, a_i) + c_i^0(\lambda_s)(1 - h(x_i, a_i))$$

Writing $L_i(\lambda_s) = c_i^1(\lambda_s) - c_i^0(\lambda_s)$, the objective becomes $\sum_{i=1}^n L_i(\lambda_s) h(x_i, a_i)$. Hence, $\nabla_{h_s} U(h_s, \lambda_s)$ is linear in $h_s$ and equals the vector $\{L_i(\lambda_s)\}_{i=1}^n$. With this observation, the objective in Equation (7) becomes

$$\eta \sum_{s=1}^t \sum_{i=1}^n L(\lambda_s) h(x_i, a_i) + \frac{1}{2} \sum_{i=1}^n (h(x_i, a_i))^2$$

$$\leq \eta \sum_{i=1}^n L\left(\sum_{s=1}^t \lambda_s\right) h(x_i, a_i) + \frac{1}{2} \sum_{i=1}^n h(x_i, a_i) = \sum_{i=1}^n \left(\eta L\left(\sum_{s=1}^t \lambda_s\right) + \frac{1}{2}\right) h(x_i, a_i).$$

The first inequality follows from two observations – $L(\lambda)$ is linear in $\lambda$, and, since the predictions $h(x_i, a_i) \in [0, 1]$ we replace the quadratic term by a linear term, an upper bound.[3]

Finally, we observe that even though the number of weights in $\mathcal{N}(\gamma_1, \mathcal{W})$ is exponential in $n$, Algorithm 3 can be efficiently implemented. This is because the best response of the $\lambda$-player always returns a solution where all the entries are zero or exactly one is set to $B$. Therefore, instead of recording the entire $\lambda$ vector the algorithm can just record the non-zero variables and there will be at most $T$ of them. The next lemma provides performance guarantees of Algorithm 3.

**Theorem 2.** *Given a desired fairness level $\varepsilon$, if Algorithm 3 is run for $T = O\left(\frac{n}{\varepsilon^2}\right)$ rounds, then the ensemble hypothesis $\widehat{h}$ provides the following guarantee:*

$$\sum_{i=1}^n w_i^0 \ell(\widehat{h}(x_i, a_i), y_i) \leq \min_{h \in \mathcal{H}} \sum_{i=1}^n w_i^0 \ell(h(x_i, a_i), y_i) + O(\varepsilon) \text{ and } \delta_{DP}^w(\widehat{h}) \leq 2\varepsilon \quad \forall w \in \mathcal{W}.$$

**ALGORITHM 3:** Approximate Fair Classifier (ApxFair)

---

**Input:** $\eta > 0$, weight $w^0 \in \mathbb{R}^n_+$, number of rounds $T$
Set $h_1 = 0$
**for** $t \in [T]$ **do**
$\quad$ $\lambda_t = \text{Best}_\lambda(h_t)$
$\quad$ Set $\widetilde{\lambda}_t = \sum_{t'=1}^t \lambda_{t'}$
$\quad$ $h_{t+1} = \arg\min_{h \in \mathcal{H}} \sum_{i=1}^n (\eta L_i(\widetilde{\lambda}_t) + 1/2)h(x_i, a_i)$
**end**
**return** *Uniform distribution over* $\{h_1, \ldots, h_T\}$.

---

## 4 Experiments

We used the following four datasets for our experiments.

- **Adult.** In this dataset [24], each example represents an adult individual, the outcome variable is whether that individual makes over $50k a year, and the protected attribute is gender. We work with a balanced and preprocessed version with 2,020 examples and 98 features, selected from the original 48,882 examples.

- **Communities and Crime.** In this dataset from the UCI repository [29], each example represents a community. The outcome variable is whether the community has a violent crime rate in the 70th percentile of all communities, and the protected attribute is whether the community has a majority white population. We used the full dataset of 1,994 examples and 123 features.

- **Law School.** Here each example represents a law student, the outcome variable is whether the law student passed the bar exam or not, and the protected attribute is race (white or not white). We used a preprocessed and balanced subset with 1,823 examples and 17 features [33].

- **COMPAS.** In this dataset, each example represents a defendant. The outcome variable is whether a certain individual will recidivate, and the protected attribute is race (white or black). We used a 2,000 example sample from the full dataset.

For Adult, Communities and Crime, and Law School we used the preprocessed versions found in the accompanying GitHub repo of [22][4]. For COMPAS, we used a sample from the original dataset [1].

In order to evaluate different fair classifiers, we first split each dataset into five different random 80%-20% train-test splits. Then, we split each training set further into a 80%-20% train and validation sets. Therefore, there were five random sets of 64%-16%-20% train-validation-test split. For each split, we used the validation set to select the hyperparameters, train set to build a model, and the test set to evaluate its performance (fairness and accuracy). Finally we aggregated these metrics across the five different test sets to obtain average performance.

We compared our algorithm to: a pre-processing method of [20], an in-processing method of [4], a post-processing method of [19]. For our algorithm [5], we use scikit-learn's logistic regression [27] as the learning algorithm in Algorithm 3. We also show the performance of unconstrained logistic regression. To find the correct hyper-parameters ($B, \eta, T,$ and $T_m$) for our algorithm, we fixed $T = 10$ for EO, and $T = 5$ for DP, and used grid search for the hyper-parameters $B, \eta,$ and $T_m$. The tested values were $\{0.1, 0, 2, \ldots, 1\}$ for $B$, $\{0, 0.05, \ldots, 1\}$ for $\eta$, and $\{100, 200, \ldots, 2000\}$ for $T_m$.

**Results.** We computed the maximum violating weight by solving a LP that is the same as the one used by the best response oracle (Algorithm 2), except that we restrict individual weights to be in the range $[(1-\epsilon)/n, (1+\epsilon)/n]$, and keep protected group marginals the same. This keeps the $\ell_1$ distance between weighted and unweighted distributions within $\epsilon$. Figure 2 compares the robustness of our classifier against the other fair classifiers, and we see that for both DP and EO, the fairness violation of our classifier grows more slowly as $\epsilon$ increases, compared to the others, suggesting robustness to $\epsilon$-perturbations of the distribution. Our algorithm also performs comparatively well in both accuracy and fairness violation to the existing fair classifiers, though there is a trade-off between robustness and test accuracy. The unweighted test accuracy of our algorithm drops by at most 5%-10% on all

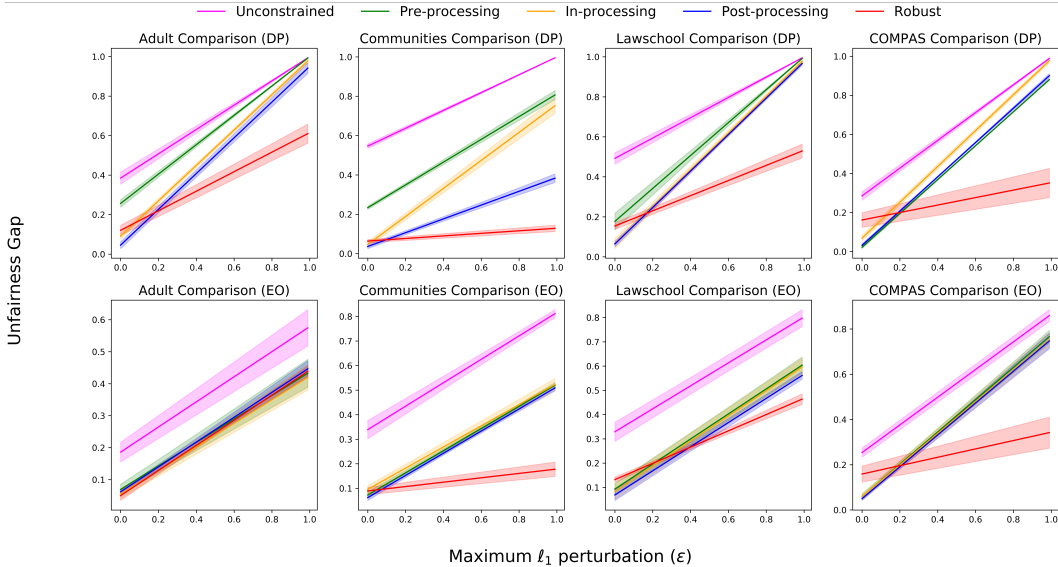

Figure 2: *DP and EO Comparison.* We vary the $\ell_1$ distance $\epsilon$ on the x-axis and plot the fairness violation on the y-axis. We use five random 80%-20% train-test splits to evaluate test accuracy and fairness. The bands across each line show standard error. For both DP and EO fairness, our algorithm is significantly more robust to reweightings that are within $\ell_1$ distance $\epsilon$ on most datasets.

datasets, suggesting that robustness comes at the expense of test accuracy on the original distribution. However, on the test set (which is typically obtained from the same source as the original training data), the difference in fairness violation between our method and other methods is almost negligible on all the datasets, except for the COMPAS dataset, where the difference it at most 12%. See the supplementary material for full details of this trade-off.

# 5   Conclusion and Future Work

In this work, we study the design of fair classifiers that are robust to weighted perturbations of the dataset. An immediate future work is to consider robustness against a broader class of distributions like the set of distributions with a bounded $f$-divergence or Wasserstein distance from the training distribution. We also considered statistical notions of fairness and it would be interesting to perform a similar fairness vs robustness analysis for other notions of fairness.

# 6   Broader Impact

This paper addresses a topic of societal interest that has received considerable attention in the NeurIPS community over the past several years. Two anticipated positive outcomes of our work include: (i) increased awareness of the dataset robustness concerns when evaluating fairness criteria, and (ii) algorithmic techniques for coping with these robustness concerns. Thus, we believe researchers and practitioners who seek to develop classifiers that satisfy certain fairness criteria would be the primary beneficiaries of this work. A possible negative outcome of the work is the use of the techniques to declare some classifiers as "fair" without proper consideration of the semantics of the fairness criteria or the underlying dataset used to evaluate these criteria. Thus, individuals subject to the decisions made by such classifiers could be adversely affected. Finally, our experimental evaluations use "real world" datasets, so the conclusions that we draw in our paper may have implications for individuals associated with these data.

**Acknowledgments**: We thank Shipra Agrawal and Roxana Geambasu for helpful preliminary discussions. DM was supported through a Columbia Data Science Institute Post-Doctoral Fellowship. DH was partially supported by NSF awards CCF-1740833 and IIS-15-63785 as well as a Sloan Fellowship. SJ was partially supported by NSF award CNS-18-01426.

## Footnotes

[1]We consider the average of false positive rate and true positive rate for simplicity, and our method can also handle more general definitions of EO. [28]

[2]All the omitted proofs are provided in the supplementary material.

[3]Without this relaxation we will have to solve a regularized version of cost-sensitive classification.

[4]https://github.com/algowatchpenn/GerryFair

[5]Our code is available at this GitHub repo: https://github.com/essdeee/Ensuring-Fairness-Beyond-the-Training-Data.

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
