[Supplementary Material]

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

# A Appendix

## A.1 Maximum violating weight linear program

In our experiments, we use the following linear program to find the maximum fairness violating weighted distribution, while keeping individual weights to the range $[(1-\epsilon)/n, (1+\epsilon)/n]$, and keep protected group marginals the same:

$$\max_w \frac{1}{\pi_a} \sum_{i:a_i=a} w_i h(x_i, a) - \frac{1}{\pi'_a} \sum_{i:a_i=a'} w_i h(x_i, a')$$

$$\text{s.t.} \sum_{i:a_i=a} w_i = \pi_a \ \forall a \in \mathcal{A}$$

$$\frac{1-\epsilon}{n} \leq w_i \leq \frac{1+\epsilon}{n} \ \forall i \in [n]$$

$$\sum_i w_i = 1.$$

Here, the $\pi_a$ are the original protected group marginal probabilities.

## A.2 Proof of Theorem 1

The proof of this theorem is similar to the proof of Theorem 7 in [12] except that we use additive approximate oracle. Let $v^* = \min_{h \in \mathcal{H}_\mathcal{W}} \max_{w \in \mathcal{W}} \ell(h, w)$. Recall that the $w$-player plays projected gradient descent algorithm, whereas the $h$-player uses ApxFair$(\cdot)$ to generate $\alpha$-approximate best response. By the guarantee of the projected gradient descent algorithm, we have

$$\frac{1}{T_m} \sum_{t=1}^{T_m} \ell(h_t, w_t) \geq \max_{w \in \mathcal{W}} \frac{1}{T_m} \sum_{t=1}^{T_m} \ell(h_t, w) - \max_{w \in \mathcal{W}} \|w\|_2 \sqrt{\frac{2}{T_m}}$$

$$\geq \max_{w \in \mathcal{W}} \frac{1}{T_m} \sum_{t=1}^{T_m} \ell(h_t, w) - \sqrt{\frac{2}{T_m}}$$

The last inequality follows because the weights always sum to one, so $\|w\|_2 \leq \sqrt{\|w\|_1} \leq 1$.

$$v^* = \min_{h \in \mathcal{H}_\mathcal{W}} \max_{w \in \mathcal{W}} \ell(h, w) \geq \min_{h \in \mathcal{H}_\mathcal{W}} \frac{1}{T_m} \sum_{t=1}^{T_m} \ell(h, w_t) \geq \frac{1}{T_m} \sum_{t=1}^{T_m} \min_{h \in \mathcal{H}_\mathcal{W}} \ell(h, w_t)$$

$$\geq \frac{1}{T_m} \left( \sum_{t=1}^{T_m} \ell(h_t, w_t) - \alpha \right) = \frac{1}{T_m} \sum_{t=1}^{T_m} \ell(h_t, w_t) - \alpha$$

$$\geq \max_{w \in \mathcal{W}} \frac{1}{T_m} \sum_{t=1}^{T_m} \ell(h_t, w) - \sqrt{\frac{2}{T_m}} - \alpha$$

The third inequality follows from the $\alpha$-approximate fairness of ApxFair$(\cdot)$. Now rearranging the last inequality we get $\max_{w \in \mathcal{W}} \frac{1}{T_m} \sum_{t=1}^{T_m} \ell(h_t, w) \leq v^* + \sqrt{2/T_m} + \alpha$, the desired result.

## A.3 Proof of Lemma 1

Recall the definition of demographic parity with respect to a weight vector $w$.

$$\delta_{DP}^w(f) = \left| \frac{\sum_{i:a_i=a} w_i f(x_i, a)}{\sum_{i:a_i=a} w_i} - \frac{\sum_{i:a_i=a'} w_i f(x_i, a')}{\sum_{i:a_i=a'} w_i} \right|$$

For a given weight $w$, we construct a new weight $w' = (w'_1, \ldots, w'_n)$ as follows. For each $i \in [n]$, $w'_i$ is the upper-end point of the bucket containing $w_i$. Note that this guarantees that either $w_i \leq \delta$ or $\frac{w'_i}{1+\gamma_1} \leq w_i \leq w'_i$. We now establish the following lower bound.

$$\frac{\sum_{i:a_i=a} w_i f(x_i, a)}{\sum_{i:a_i=a} w_i} \geq \frac{1}{1+\gamma_1} \frac{\sum_{i:a_i=a} w'_i f(x_i, a)}{\sum_{i:a_i=a} w'_i} \geq (1-\gamma_1) \frac{\sum_{i:a_i=a} w'_i f(x_i, a)}{\sum_{i:a_i=a} w'_i} \quad (8)$$

Also note that,

$$\sum_{i:a_i=a} w_i' \leq \sum_{i:a_i=a,w_i>\delta} w_i + \sum_{i:a_i=a,w_i\leq\delta} \delta \leq (1+\gamma_1) \sum_{i:a_i=a,w_i>\delta} w_i' + n\delta$$

This gives us the following.

$$\frac{\sum_{i:a_i=a} w_i f(x_i,a)}{\sum_{i:a_i=a} w_i} \leq \frac{\sum_{i:a_i=a} w_i' f(x_i,a)}{\frac{1}{1+\gamma_1}\sum_{i:a_i=a} w_i' - \frac{n\delta}{1+\gamma_1}} \leq (1+\gamma_1)\frac{\sum_{i:a_i=a} w_i' f(x_i,a)}{\sum_{i:a_i=a} w_i' - n\delta}$$

Now we substitute, $\delta = \gamma_1/(2n)$ and get the following upper bound.

$$\frac{\sum_{i:a_i=a} w_i f(x_i,a)}{\sum_{i:a_i=a} w_i} \leq (1+\gamma_1)\frac{\sum_{i:a_i=a} w_i' f(x_i,a)}{\sum_{i:a_i=a} w_i' - \gamma_1/2}$$

$$\leq \frac{1+\gamma_1}{1-\gamma_1}\frac{\sum_{i:a_i=a} w_i' f(x_i,a)}{\sum_{i:a_i=a} w_i'} \leq (1+3\gamma_1)\frac{\sum_{i:a_i=a} w_i' f(x_i,a)}{\sum_{i:a_i=a} w_i'} \qquad (9)$$

Now we bound $\delta_{DP}^w(f)$ using the results above. Suppose $\frac{\sum_{i:a_i=a} w_i f(x_i,a)}{\sum_{i:a_i=a} w_i} > \frac{\sum_{i:a_i=a'} w_i f(x_i,a')}{\sum_{i:a_i=a'} w_i}$. Then we have,

$$\delta_{DP}^w(f) \leq (1+3\gamma_1)\frac{\sum_{i:a_i=a} w_i' f(x_i,a)}{\sum_{i:a_i=a} w_i'} - (1-\gamma_1)\frac{\sum_{i:a_i=a} w_i' f(x_i,a)}{\sum_{i:a_i=a} w_i'}$$

$$\leq \frac{\sum_{i:a_i=a} w_i' f(x_i,a)}{\sum_{i:a_i=a} w_i'} - \frac{\sum_{i:a_i=a} w_i' f(x_i,a)}{\sum_{i:a_i=a} w_i'} + 4\gamma_1$$

$$\leq \delta_{DP}^{w'}(f) + 4\gamma_1$$

The first inequality uses the upper bound for the first term (Eq. (9)) and the lower bound for the second term (Eq. (8)). The proof when the first term is less than the second term in the definition of $\delta_{DP}^w(f)$ is similar. Therefore, if we guarantee that $\delta_{DP}^{w'}(f) \leq \varepsilon$, we have $\delta_{DP}^w(f) \leq \varepsilon + 4\gamma_1$.

### A.4 Proof of Lemma 2

We need to consider two cases. First, suppose that $R(w,a,h) - R(w,a',h) \leq \varepsilon - 4\gamma_1$ for all $w \in \mathcal{N}(\gamma_1,\mathcal{W})$ and $a,a' \in \mathcal{A}$. Then, $\delta_{DP}^w(h) = \sup_{a,a'\in\mathcal{A}} |R(w,a,h) - R(w,a',h)| \leq \varepsilon - 4\gamma_1$ for any weight $w \in \mathcal{N}(\gamma_1,\mathcal{W})$. Therefore, by Lemma 1, for any weight $w \in \mathcal{W}$, we have $\delta_{DP}^w(h) \leq \varepsilon$. Now, for any marginal $\pi \in \Pi(\gamma_2,\mathcal{A})$, and $a,a'$ consider the corresponding linear program. We show that the optimal value of the LP is bounded by $\varepsilon$. Indeed, consider any weight $w$ satisfying the marginal conditions, i.e., $\sum_{i:a_i=a} w_i = \pi_a$ and $\sum_{i:a_i=a'} w_i = \pi_{a'}$. Then, the objective of the LP is

$$\frac{1}{\pi_a}\sum_{i:a_i=a} w_i h(x_i,a) - \frac{1}{\pi_{a'}}\sum_{i:a_i=a'} w_i h(x_i,a') \leq \sup_{w\in\mathcal{W}} \delta_{DP}^w(h) \leq \varepsilon.$$

This implies that the optimal value of the LP is always less than $\varepsilon$. So Algorithm 2 returns the zero vector, which is also the optimal solution in this case.

Second, there exists $w \in \mathcal{N}(\gamma_1,\mathcal{W})$ and groups $a,a'$ such that $R(w,a,h) - R(w,a',h) > \varepsilon - 4\gamma_1$ and in particular let $(w^*,a_1^*,a_2^*) \in \arg\max_{w,a,a'} T(w,a,h) - T(w,a',h)$. Then the optimal solution sets $\lambda_{w^*}^{a_1^*,a_2^*}$ to $B$ and everything else to zero. Let $\pi_{a_1^*}$ and $\pi_{a_2^*}$ be the corresponding marginals for groups $a_1^*$ and $a_2^*$, and let $\pi_{a_1^*}'$ and $\pi_{a_2^*}'$ be the upper-end points of the buckets containing $\pi_{a_1^*}$ and $\pi_{a*2}$ respectively. As $\pi_{a_1^*}$ is marginal for a weight belonging to the set $\mathcal{N}(\gamma_1,\mathcal{W})$ and any weight in $\mathcal{N}(\gamma_1,\mathcal{W})$ puts at least $2\gamma_1/n$ on any training instance, we are always guaranteed that

$$\pi_{a_1^*} \geq \frac{2\gamma_1}{n} \geq \frac{\delta}{1+\gamma_2}.$$

This guarantees the following inequalities

$$\frac{\pi_{a_1^*}'}{1+\gamma_2} \leq \pi_{a_1^*} \leq \pi_{a_1^*}'.$$

Similarly, we can show that

$$\frac{\pi'_{a_2^*}}{1 + \gamma_2} \leq \pi_{a_2^*} \leq \pi'_{a_2^*}.$$

Now, consider the LP corresponding to the marginal $\pi'$ and subgroups $a_1^*$ and $a_2^*$.

$$\frac{1}{\pi'_{a_1^*}} \sum_{i:a_i=a_1^*} w_i h(x_i, a_1^*) - \frac{1}{\pi'_{a_2^*}} \sum_{i:a_i=a_2^*} w_i h(x_i, a_2^*)$$

$$\geq \frac{1}{(1+\gamma_2)\pi_{a_1^*}} \sum_{i:a_i=a_1^*} w_i h(x_i, a_1^*) - \frac{1}{\pi_{a_2^*}} \sum_{i:a_i=a_2^*} w_i h(x_i, a_2^*)$$

$$\geq (1 - \gamma_2)R(w, a_1^*, h) - R(w, a_2^*, h)$$

$$\geq R(w, a_1^*, h) - R(w, a_2^*, h) - \gamma_2$$

Therefore, if the maximum value of $R(w, a, h) - R(w, a', h)$ over all weights $w$ and subgroups $a, a'$ is larger than $\varepsilon + \gamma_2$, the value of the corresponding LP will be larger than $\varepsilon$ and the algorithm will set the correct coordinate of $\lambda$ to $B$. On the other hand, if the maximum value of $R(w, a, h) - R(w, a', h)$ is between $\varepsilon - 4\gamma_1$ and $\varepsilon + \gamma_2$. In that case, the algorithm might return the zero vector with value zero. However, the optimal value in that case can be as large as $B \times (4\gamma_1 + \gamma_2)$.

### A.5 Proof of Theorem 2

We first recall the following guarantee about the performance of the RFTL algorithm.

**Lemma 3** (Restated Theorem 5.6 from [21]). *The RFTL algorithm achieves the following regret bound for any $u \in \mathcal{K}$*

$$\sum_{t=1}^{T} f_t(x_t) - f_t(u) \leq \frac{\eta}{4} \sum_{t=1}^{T} \|\nabla f_t(x_t)\|_\infty^2 + \frac{R(u) - R(x_1)}{2\eta}$$

*Moreover, if $\|\nabla f_t(x_t)\|_\infty \leq G_R$ for all $t$ and $R(u) - R(x_1) \leq D_R$ for all $u \in \mathcal{K}$, then we can optimize $\eta$ to get the following bound: $\sum_{t=1}^{T} f_t(x_t) - f_t(u) \leq D_R G_R \sqrt{T}$.*

The statement of this theorem follows from two lemmas. Lemma 4 proves that if Algorithm 3 is run for $T$ rounds, it computes a $(2M + B)\sqrt{n/T} + B(4\gamma_1 + \gamma_2)$-minmax equilibrium of the game $U(h, \lambda)$. On the other hand, Lemma 5 proves that any $\nu$-approximate solution $(\widehat{h}, \widehat{\lambda})$ of the game $U(h, \lambda)$ has two properties

1. $\widehat{h}$ minimizes training loss with respect to the weight $w^0$ up to an additive error of $2\nu$.
2. $\widehat{h}$ provides $\varepsilon$-fairness guarantee with respect to the set of all weights in $\mathcal{W}$ upto an additive error fo $\frac{M+2\nu}{B}$.

Now substituting $\nu = (2M + B)\sqrt{n/T} + B(4\gamma_1 + \gamma_2)$ we get the following two guarantees:

$$\sum_{i=1}^{n} w_i^0 \ell(\widehat{h}(x_i, a_i), y_i) \leq \min_{h \in \mathcal{H}} \sum_{i=1}^{n} w_i^0 \ell(h(x_i, a_i), y_i) + 2(2M + B)\sqrt{\frac{n}{T}} + 2B(4\gamma_1 + \gamma_2)$$

and

$$\forall w \in \mathcal{W} \quad \delta_{DP}^w(\widehat{h}) \leq \varepsilon + \frac{M}{B} + 2(4\gamma_1 + \gamma_2) + \left(1 + \frac{2M}{B}\right)\sqrt{\frac{n}{T}}.$$

Now we can set the following values for the parameters $B = 3M/\varepsilon$, $T = 36n/\varepsilon^2$, $4\gamma_1 + \gamma_2 = \varepsilon/6$, and get the desired result.

**Lemma 4.** *Suppose $|\ell(y, \widehat{y})| \leq M$ for all $y, \widehat{y}$. Then Algorithm 3 computes a $(2M + B)\sqrt{n/T} + B(4\gamma_1 + \gamma_2)$-approximate minmax equilibrium of the game $U(h, \lambda)$ for $h \in \mathcal{H}$ and $\lambda \in \mathbb{R}_+^{|N(\gamma_1, \mathcal{W})| \times |\mathcal{A}|^2}, \|\lambda\|_1 \leq B$.*

*Proof.* At round $t$, the cost is linear in $h_t$, i.e., $f_t(h_t) = \sum_{i=1}^{n} L(\lambda_t)_i h_t(x_i, a_i)$. Let us write $\bar{\lambda} = \frac{1}{T}\lambda_t$ and $D$ to be the uniform distribution over $h_1, \ldots, h_T$. Since we chose $R(x) = \frac{1}{2}\|x\|_2^2$ as

the regularization function and the actions are $[0,1]$ vectors in $n$-dimensional space, the diameter $D_R$ is bounded by $\sqrt{n}$. On the other hand, $\|\nabla f_t(h_t)\|_\infty = \max_i |L(\lambda_t)_i|$. We now bound $|L(\lambda_t)_i|$ for an arbitrary $i$. Suppose $y_i = 1$. The proof when $y = 0$ is identical.

$$|L(\lambda_t)_i| = \left|c_i^1 - c_i^0\right| = \left|w_i^0\right| |\ell(0,1) - \ell(1,1)| + |\Delta_i|$$
$$\leq 2M + B$$

The last line follows as $w_i^0 \leq 1$ and since $\lambda_t$ is an approximate best reponse computed by Algorithm 2, exactly one $\lambda$ variable is set to $B$. Therefore, by Theorem 3, for any hypothesis $h \in \mathcal{H}$,

$$\sum_{t=1}^{T}\sum_{i=1}^{n} L(\lambda_t)_i h_t(x_i, a_i) - \sum_{i=1}^{n} L(\lambda_t)_i h(x_i, a_i) \leq (2M+B)\sqrt{nT}$$

$$\Leftrightarrow \sum_{t=1}^{T} U(h_t, \lambda_t) - U(h, \lambda_t) \leq (2M+B)\sqrt{nT}$$

$$\Leftrightarrow \frac{1}{T}\sum_{t=1}^{T} U(h_t, \lambda_t) \leq U(h, \bar{\lambda}) + \frac{(2M+B)\sqrt{n}}{\sqrt{T}} \tag{10}$$

On the other hand, $\lambda_t$ is an approximate $B(4\gamma_1 + \gamma_2)$-approximate best response to $h_t$ for each round $t$. Therefore, for any $\lambda$ we have,

$$\sum_{t=1}^{T} U(h_t, \lambda_t) \geq \sum_{t=1}^{T} U(h_t, \lambda) - BT(4\gamma_1 + \gamma_2)$$

$$\Leftrightarrow \frac{1}{T}\sum_{t=1}^{T} U(h_t, \lambda_t) \geq \mathbb{E}_{h\sim D} U(h, \lambda) - B(4\gamma_1 + \gamma_2) \tag{11}$$

Equations (10) and (11) immediately imply that the distribution $D$ and $\bar{\lambda}$ is a $(2M+B)\sqrt{n/T} + B(4\gamma_1 + \gamma_2)$-approximate equilibrium of the game $U(h, \lambda)$ [16]. $\qed$

**Lemma 5.** *Let $(\widehat{h}, \widehat{\lambda})$ be a $\nu$-approximate minmax equilibrium of the game $U(h, \lambda)$. Then,*

$$\sum_{i=1}^{n} w_i^0 \ell(\widehat{h}(x_i, a_i), y_i) \leq \min_{h\in\mathcal{H}} \sum_{i=1}^{n} w_i^0 \ell(h(x_i, a_i), y_i) + 2\nu$$

*and*

$$\forall w \in \mathcal{W} \quad \delta_{DP}^w(\widehat{h}) \leq \varepsilon + \frac{M + 2\nu}{B}$$

*Proof.* Let $(\widehat{h}, \widehat{\lambda})$ be a $\nu$-approximate minmax equilibrium of the game $U(h, \lambda)$, i.e.,

$$\forall h \quad U(\widehat{h}, \widehat{\lambda}) \leq U(h, \widehat{\lambda}) + \nu \quad \text{and} \quad \forall \lambda \quad U(\widehat{h}, \widehat{\lambda}) \geq U(\widehat{h}, \lambda) - \nu$$

Let $h^*$ be the optimal feasible hypothesis. First suppose that $\widehat{h}$ is feasible, i.e., $T(w, a, \widehat{h}) - T(w, a', \widehat{h}) \leq \varepsilon - 4\gamma_1$ for all $w \in N(\gamma_1, \mathcal{W})$ and $a, a' \in \mathcal{A}$. In that case, the optimal $\lambda$ is the zero vector and $\max_\lambda U(\widehat{h}, \lambda) = \sum_{i=1}^{n} w_i^0 \ell(h(x_i, a_i), y_i)$. Therefore,

$$\sum_{i=1}^{n} w_i^0 \ell(\widehat{h}(x_i, a_i), y_i) = \max_\lambda U(\widehat{h}, \lambda) \leq U(\widehat{h}, \widehat{\lambda}) + \nu \leq U(h^*, \widehat{\lambda}) + 2\nu \leq \sum_{i=1}^{n} w_i^0 \ell(h^*(x_i, a_i), y_i) + 2\nu$$

The last inequality follows because $h^*$ is feasible and $\lambda$ is non-negative. Now consider the case when $\widehat{h}$ is not feasible, i.e., there exists $w, a, a'$ such that $T(w, a, \widehat{h}) - T(w, a', \widehat{h}) > \varepsilon - 4\gamma_1$. In that case, let $(\widehat{w}, \widehat{a}, \widehat{a}')$ be the tuple with maximum violation and the optimal $\lambda$, say $\lambda^*$, sets this coordinate to $B$ and everything else to zero. Then

$$\sum_{i=1}^{n} w_i^0 \ell(\widehat{h}(x_i, a_i), y_i) = U(\widehat{h}, \lambda^*) - B(T(\widehat{w}, \widehat{a}, \widehat{h}) - T(\widehat{w}, \widehat{a}', \widehat{h}) - \varepsilon + 4\gamma_1)$$

$$\leq U(\widehat{h}, \lambda^*) \leq U(\widehat{h}, \widehat{\lambda}) + \nu \leq U(h^*, \widehat{\lambda}) + 2\nu \leq \sum_{i=1}^{n} w_i^0 \ell(h^*(x_i, a_i), y_i) + 2\nu.$$

Figure 3: *Fairness v. Accuracy.* We plot the accuracy (x-axis) vs. the fairness violation (y-axis) for demographic parity and equalized odds for our robust and fair classifier. The reported values are averages over five random 80%-20% train-test splits, with standard error bars. We observe that the fairness violation is mostly comparable to existing state-of-the-art fair classifiers, though robustness comes at the expense of somewhat lower test accuracy.

The previous chain of inequalities also give

$$B\left(\max_{(w,a,a')} T(w,a,\widehat{h}) - T(w,a',\widehat{h}) - \varepsilon + 4\gamma_1\right) \leq \sum_{i=1}^{n} w_i^0 \ell(h^*(x_i,a_i),y_i) + 2\nu \leq M + 2\nu.$$

This implies that for all weights $w \in N(\gamma_1, \mathcal{W})$ the maximum violation of the fairness constraint is $(M + 2\nu)/B$, which in turn implies a bound of at most $(M + 2\nu)/B + \varepsilon$ on the fairness constraint with respect to any weight $w \in \mathcal{W}$. $\qquad\square$

# B   Fairness vs. Accuracy Tradeoff

In Figure 3, we see the accuracy and fairness violation of our algorithm against the other state-of-the-art fair classifiers. We find that, though the fairness violation is mostly competitive with the existing fair classifiers, robustness against weighted perturbations comes at the expense of somewhat lower test accuracy.