[Reviews · NeurIPS 2020]

Review 1

Summary and Contributions: This paper tackles the algorithmic fairness problem with the goal to make it more robust. The main idea is to have a fair classifier not only on the specific training distribution at hand, but w.r.t. a family of distributions obtained by perturbation (weighting) the original training examples.

Strengths: - Importance of the topic in the area of algorithmic fairness - Clarity of the problem tackled and the proposed solution - Theoretical results

Weaknesses: - The description of the experiments is not complete in the main paper, making reproducibility almost impossible and also difficult to evaluate the statistical relevance of the results. Reading the supplementary material (section B), some details are there more clearly stated (e.g. the split 80% for training and 20% for test, and the fact there are 5 different splits). Unfortunately, it is still not clear even reading the supplementary material: (a) what is the validation accuracy and fairness the authors refers to? (The split is between train 80% and test 20%, so which is the “validation” set?). Why the authors fixed “by hand” some of the hyper-parameters? This missing details make the experimental results difficult replicate and also to evaluate correctly.

Correctness: - The method looks correct and well described - Good theory part - The experimental section is missing key details

Clarity: - The problem is clearly described and the solution is well motivated - The method is straightforward and it sounds - The experiments are not well exposed (see Weaknesses section)

Relation to Prior Work: The contributions are clearly stated in the paper and the correct references are reported

Reproducibility: No

Additional Feedback: A better description of the experiments -- especially concerning validation and hyper-parameters selection -- is needed to make the results relevant and to understand the benefit of using this new proposed method After the rebuttal: I increased my vote thanks to the explanation in the rebuttal. I suggest to the authors to add all the experimental details (about validation and hyper-parameters) in the main paper, because they are of key importance for the validity and reproducibility of the full experimental section.


Review 2

Summary and Contributions: The paper discusses the design of fair, loss-minimizing classifiers that are robust to reweightings of the training data. They discuss a scheme where they nest two optimization processes. The outer process is a robust loss minimizer. The inner process deals with the fairness constraint. They have experiments for real world data sets.

Strengths: The paper is a good mix of theory with practical results. I am not an expert on optimization, so I am not calibrated to speak to the technical novelty. But in the very least the paper is a very nice composition of existing ideas with practical results to match.

Weaknesses: I wish the paper spent more time discussing Figure 1 and the Experiments. I think it would have been more valuable to describe the main technical challenges and how they were circumvented and spend a little less space on the formulae. (Thank you for the response!)

Correctness: To the best of my knowledge.

Clarity: Yes.

Relation to Prior Work: Yes.

Reproducibility: Yes

Additional Feedback:


Review 3

Summary and Contributions: The focus area of the paper is fairness-aware classification. However, the paper goes beyond the existing techniques and proposes methods to train fair classifiers that are distributionally robust -- where the classifier is fair not only w.r.t. the distribution of the observed data, but also fair w.r.t some neighboring distributions. Neighboring distributions are characterized using different weighted sums of training data points. The paper then casts the problem as a min-max program where an adversary tries to find the most unfair weights whereas the learner aims to learn the fair classifier that is optimal w.r.t. these weights. Experiments on real world datasets show that the proposed method is more robust to distributional shift than existing fair learning methods.

Strengths: 1- The problem that the paper attempts has a very solid motivation -- training fair models that also account for distributional shift. Given the fact that the real world dataset likely contain measurement errors and biases (e.g., sampling bias from different groups), it is indeed quite important to ensure that the learning methods are distributionally robust.

Weaknesses: 1- The biggest point where the paper can be improved is connecting the definition of "robustness" to real world biases and measurement errors. Specifically, the paper lays out the motivation quite nicely in the paragraph starting line 24 by stating that the real world datasets, specially the ones in the fairness domain, are prone to measurement noise and biases. However, it is not clearly stated how the definition of robustness, which follows from the weight-based characterization of neighboring distributions, relates to these issues with the real world data. For instance, how does the weighted sample based definition related to attribute or label measurement error? In this reviewer's opinion, this connection with the real world is quite important for a paper attempting an issue such as fairness. Without these connections, the utility of the methods in the real world might be limited. 2- The writing of the paper can be improved at several places. Specifically, the notation is a bit confusing at some points. Also, the experiments seem to be missing important discussion and details. Please see points 1-4 in the "Additional Feedback" section below.

Correctness: Seems to be the case indeed. However, some guidelines on selecting hyperparameter ranges should be included in the paper.

Clarity: Notation can be improved. Please see Additional feedback for details.

Relation to Prior Work: The relation of the paper with the prior work on fairness-aware learning is quite clear. The paper borrows ideas from prior work on DRO, but the differences are discussed.

Reproducibility: Yes

Additional Feedback: 1- Perhaps the reviewer missed it, but was is "f" in Eq (1) onwards? Is f(x,a) the same as h(x,a) -- which is defined in line 80? 2- The statement in the text immediately after Eq (1) seems incorrect. DP is supposed to be the difference in "acceptance rate" of the two groups a and a'. This would also make sense assuming that f() returns the predicted label. 3- The paper seems to skim over very important experimental details. For instance, some discussion is needed into the drop in accuracy for the Robust classifier. The drop seems too high for Adult and COMPAS datasets (these are the datasets that this reviewer knows). The accuracy of Robust (Fig 3 in appendix) on these datasets seems to be close to, or even lower than the accuracy of a classifier that always predicts the majority class. Perhaps it would help to also show the AUC in addition to accuracy? 4- Similarly, it would help to add a baseline which corresponds to a "Robust but unfair" classifier. This might serve as a nice reference point to the proposed method and add some context to the drop in accuracy. 5- How much additional run time does the method require as compared to the vanilla (unfair) classifier? Similarly, comparing the runtime to other fair baselines would be very useful. 6- In line 80, h(x,a) requires the protected attribute as well. How (if at all) do the results change if h() only operates on x -- that is, if a is not available at test time? = POST REBUTTAL COMMENTS = Thanks very much to the authors for the helpful response. Most of my questions were answered, and I am increasing my score as a result from 5 to 6. My biggest concern was regarding how the reweighing of the data corresponds to biased data collection. The part about under-representation was answered in a fairly satisfactory way. However, I am not sure about the measurement error part. Precisely, regarding line 31 in the rebuttal, I do not think that all the measurement errors will be such that the feature values are under-reported. the last sentence ("More general type of errors.."), it is not clear what "assigning weights to each individual and feature pair" means. What are the feature pairs here? Does reweighing individual features (rather than whole data points) affect the overall algorithm? The paper doesn't have to solve all the issues related to bias in data collection, and it is OK to mention that it covers only parts of the problem (under-representation) and does not cover other parts (measurement errors). Also, regarding h(x,a) and h(x), of course the sensitive feature is needed to audit the outcomes for fairness. However, the question was, whether h() necessarily need the sensitive feature to output its decisions. Would be great if this was clarified in the final version.


Review 4

Summary and Contributions: The paper aims at an algorithm that achieves approximate-fairness while being robust to perturbations in the data distribution (that is, having the best performance among all perturbations, while approximately achieving the fairness constraint). Formulated as a min-max optimisation problem the authors proposes a solution scheme based on a reduction to online-learning and evaluate the empirical dependence of their algorithm on the meta-parameters of the problem.

Strengths: The paper formulates an interesting and relevant setting and proposes an algorithm that can be applied for different fairness notions. 

Weaknesses: The paper lack in clarity, mainly in Section 2: The definition of demographic parity fairness is described as discrepancy in accuracy. This is wrong as the true labels are not part of the definition. In that fairness definitions f is used while h is used elsewhere in the section for a hypothesis class element. The loss function is very limited (a discrete domain, of size 4) and consequently the hypothesis elements are confined to discrete outputs - this actually rules out most popular models and losses and renders the subsequent theoretical analysis (e.g., Theorem 1 requiring convexity of the loss function) irrelevant. The paragraph in lines 98-101 seems to require much more elaboration to be understood at this stage, and the statements made (required randomisation and convexity) are only relevant after the nature of the proposed algorithm is (much later) described.

Correctness: Due to the problem in the definition of fairness (ignoring the labels) and the limited loss-function space (discrete domain) the relevance of Theorem 1 (requiring convexity of the loss function) is questionable and it is hard to assess the validity of the proposed algorithm.

Clarity: The algorithm description is hard to follow, and its validity hard to asses given the problem in the definition of fairness (ignoring the true labels while describing it as discrepancy in accuracy). The paper could benefit tremendously if the algorithm is first described 'high-level' (expanding section 3.1) where the methods of ApxFair (and the incorporated Best_Gamma) are explained high level (but in some detail nevertheless, beyond merely their names). Again, the paragraph in lines 98-101 is misplaced or requires much more elaboration at this stage. 

Relation to Prior Work: Yes

Reproducibility: No

Additional Feedback: notation - line 93 - should be W(epsilon) Algorithm 1:  should have epsilon as an input, the subscript m in T_m is not used elsewhere. line 113: descent should be ascent (since the w-player in (2) maximizes..) ?! line 119: the proof.. (where? in the appendix?) line 105: missing 'to' Is Algorithm 2 the one called from Algorithm 3 (Best_Gamma(h_t)? It should be reflected in the name/title, etc'.  Post rebuttal: After reading the author's reply I maintain that clarity of presentation of the algorithms preclude an increased score.

[Author Response · NeurIPS 2020]

We thank all the reviewers for their feedback. They are really helpful and will improve the presentation of the paper.
We first address the common concerns, and then consider the remaining concerns of each reviewer separately.

• **Experimental Details**: We tried to include most of the important details of the experiment in the main text. We
decided to leave out details of the datasets, and hyperparameters from the main text mainly because we view our paper
providing a theoretical and algorithmic contribution. However, we realize that the choice of the hyperparameters is
important to make our work reproducible, and will include them in the main text in the next version.

• **Hyperparameter Selection**: We first split each dataset into 5 random 80%-20% training and test sets. Then, for the
experiments, we split each training set further into a 80%-20% train and validation sets (so there were 5 random sets of
64%-16%-20% train-validation-test). We used the validation sets to select the hyperparameters by performing a grid
search over the parameter space. Finally, all the reported accuracies and unfairness gaps (figures 2 and 3) correspond
to performance on averages from the 5 original test sets (which were untouched in selecting the hyperparameters). We
will update the paper to make the distinction between validation and test sets clear.

• **Definition and Confusion between $f$ and $h$**: Thanks for pointing out that we are using function $f$ instead of $h$ in
lines 87-92. In fact, they are the same and we will change it to $h$ to make this paragraph consistent with other parts
of section 2. There is a mistake on the line following eq. (1) and it should read difference in *acceptance* instead
of difference of *accuracy*. This means that the definition of demographic parity should just compare the weighted
predictions of labels between the two groups and the true label is not a part of the definition. However, the definition
of equalized odds do require true labels, as we mention in line 90.

**R1**: *Re. Validation accuracy and fairness*: validation accuracy and fairness refer to performance on the validation set
and they were used to determine the hyperparameters. The reported accuracy and fairness (figures 2 and 3) correspond
to performance on the test set, not on the validation set.

*Re. Hyper-Parameter Selection*: We fixed the number of iterations of algorithm 1 to be 10 for EO and 5 for DP.
By theorem 1, increasing this parameter will only increase the accuracy of the final randomized classifier. All the
other hyperparameters were chosen by performing grid search using the validation set. The tested values were
$\{0.1, 0.2, \ldots, 1\}$ for $B$, $\{0, 0.05, \ldots, 1\}$ for $\eta$, and $\{100, 200, \ldots, 2000\}$ for $T$.

**R2**: We agree that it is a nice idea to highlight the technical challenges of the main algorithm, and remove some of the
details of the method. This would also allow us to provide more details of the experiment in the main text.

**R4**: *Re. Weight-Based Characterization of Neighboring Distributions*: Our weight-based characterization can handle
sampling biases like under-representation of a group. This can be done by up-weighting the instances from the minority
group. In terms of attribute measurement errors, it can handle the situation when errors across all the attributes are
similar e.g. they are all under-reported. More general type of attribute errors can be handled by assigning weights to
each individual and feature pair and extending our definition for such a set of weighted distributions.

*Re. Drop in Accuracy of the Robust Classifier*: We also compared the accuracy of the optimal unfair classifier with
and without the robustness constraints. We saw a similar drop in accuracy for the ADULT dataset ($\approx 8\%$), and for the
COMPAS dataset it was even worse than what we observed for the fair classifier. This explains the drop of accuracy in
figure 3 for these two datasets. Moreover, we worked with a large set of weights $\mathcal{W}$ and if we consider a small set (e.g.
weights centered around the uniform weight with a small radius), the cost of robust classification will be lower.

*Re. Runtime*: Across the four datasets, the runtime for a full run of algorithm 1 (on a standard, 2-core 2017 Macbook
Pro laptop for $T = 10$ iterations) was 30-40 minutes across all datasets. On the other hand, the standard in-processing
based non-robust fair classifier takes about 1 minute. We didn't optimize our code and the runtime can be significantly
reduced through multiprocessing (the main bottleneck is solving the linear programs in Algorithm 2).

*Re. $h(x, a)$ vs $h(x)$*: We need access to the protected groups so that we can verify if a given classifier is unfair or not.
Without any access to the protected groups, it is impossible to estimate such biases without making strong assumptions
[Kallus et. al. FAT*-20]. In the future, we hope to extend our work for this setting under reasonable assumptions.

**R5**: *Re. Definition of Demographic Parity*: The line following eq. (1) should read difference in *acceptance* (1-
predictions) instead of difference in *accuracy*. This implies that true label is not a part of the definition of DP.

*Re. Limited Loss Function*: Fairness definitions in binary classification setting are usually considered with 0-1
predictions. This is the reason we considered loss functions over discrete domains. If one defines fairness with
real-valued predictions, we can also consider general loss functions. Moreover, for this setting, our algorithms extend
immediately – $\lambda$-best response is still given by LP, and $h$-best response still becomes a weighted classification problem.

*Re. Paragraph in lines 98-101*: We think the concept of randomized classifier and weighted ERM follow from the
discussions in the previous paragraphs. We can elaborate the non-convexity of $\mathcal{H}_\mathcal{W}$ with an example – if we use logistic
regression followed by thresholding, even simple DP fairness constraint makes the parameter space non-convex.

[Meta-Review · NeurIPS 2020]

The authors address an important area of study, aiming to ensure fairness beyond the training data by optimizing a worst case fairness loss across any weighted combination of the training set. They show that such fairness robustness comes at the cost of lower accuracy. Please add the material from the rebuttal and incorporate the reviewers' detailed comments. This includes: Add all experimental and hyperparameter details Improve clarity of writing and notation Add brief explanations of methods and proofs in the main body